# *OsOFP6* Overexpression Alters Plant Architecture, Grain Shape, and Seed Fertility

**DOI:** 10.3390/ijms25052889

**Published:** 2024-03-01

**Authors:** Xuting Zhu, Yuan Li, Xiangqian Zhao, Yukai Feng, Zhengkai Bao, Wenzhen Liu, Feifei Li

**Affiliations:** 1College of Advanced Agricultural Science, Zhejiang A&F University, Hangzhou 311300, China; zxt1477702045@163.com (X.Z.); lylylyyqq@163.com (Y.L.); zhaoxq@zafu.edu.cn (X.Z.); fengyukai0103@163.com (Y.F.); 15088793603@163.com (Z.B.); 2State Key Laboratory of Rice Biology and Breeding, China National Rice Research Institute, Hangzhou 311400, China

**Keywords:** rice, OVATE family proteins (OFPs), *OsOFP6*, mutant, plant architecture, grain shape

## Abstract

OVATE family proteins (OFPs) play important roles in plant growth and development, hormone signaling, and stress response pathways. However, the functions of OsOFPs in rice are largely unknown. In this study, a novel gain-of-function rice mutant, *Osofp6-D*, was identified. This mutant exhibited decreased plant height, erect leaves, reduced panicle size, short and wide seeds, delayed seed germination time, and reduced fertility. These phenotypic changes were attributed to the increased expression of *OsOFP6*, which was caused by a T-DNA insertion. Complementation of the *Osofp6-D* phenotype by knockout of *OsOFP6* using the CRISPR/Cas9 system confirmed that the *Osofp6-D* phenotype was caused by *OsOFP6* overexpression. In addition, transgenic plants overexpressing OsOFP6 with the 35S promoter mimicked the *Osofp6-D* phenotype. Cytological observations of the glumes showed that *OsOFP6* overexpression altered the grain shape, mainly by altering the cell shape. Hormone response experiments showed that *OsOFP6* was involved in the gibberellin (GA) and brassinolide (BR) signaling responses. Further studies revealed that OsOFP6 interacts with E3BB, which is orthologous to the *Arabidopsis* central organ size-control protein BIG BROTHER (BB). This study further elucidates the regulation mechanism of the rice OFP family on plant architecture and grain shape.

## 1. Introduction

OVATE family proteins (OFPs) are plant-specific transcription factors that play important roles in plant growth and development, hormone signaling, and stress response pathways. OFPs are characterized by a conserved C-terminal OVATE domain of approximately 70 amino acids and are widely distributed in various plants, including moss, *Arabidopsis*, rice, tomato, and maize [1]. *OVATE* was the first identified *OFP* gene to be a major quantitative trait locus (QTL) controlling pear-shaped fruit in tomatoes. A point mutation in *OVATE* leads to premature termination of translation and changes the shape of tomato fruit from round to pear-shaped [2].

There are 19 *OFP* genes in *Arabidopsis*. *AtOFP1* plays a key role in suppressing cell elongation by acting as an active transcriptional repressor. Transgenic *Arabidopsis* plants that overexpress *AtOFP1* exhibit a phenotype of reduced length in all aerial organs. A chromatin immunoprecipitation assay showed that *AtOFP1* inhibits cell elongation by suppressing the GA synthetase gene *AtGA20ox1* expression. Additionally, it regulates cotyledon development in the late embryo [3,4]. Plants overexpressing *AtOFP2*, *AtOFP4*, *AtOFP5*, and *AtOFP7* exhibited phenotypes similar to those of transgenic *Arabidopsis* overexpressing *AtOFP1*. These phenotypes included reniform cotyledons, shorter and rounder leaves, reduced fertility, and rounder seeds [1,3,4]. There are 33 *OFP* genes in rice [1]. Overexpression of *OsOFP1* and *OsOFP8* increases the leaf angle [5,6], while transgenic plants overexpressing *OsOFP3*, *OsOFP19*, and *OsOFP22* show a decreased leaf angle, reduced plant height, and altered grain shape [7,8,9]. *OsOFP2* overexpression causes reduced fertility and altered grain shape [10].

Most single and double mutant plants with the loss of function of *OFPs* do not exhibit any significant growth defects in *Arabidopsis* and rice. For example, *Atofp1*, *Atofp4*, *Atofp8*, *Atofp10*, *Atofp15*, *Atofp16*, *Osofp1*, *Atofp1Atofp4*, *Atofp15Atofp16*, and *Osofp1Osofp2* are not significantly different from their wild-type counterparts [4,6]. The *OsOFP19* RNAi strains only exhibit reduced seed thickness, while all other morphological phenotypes are not different from the wild-type [7]. It has been suggested that the functions of different OFPs are highly redundant.

In all of the reported *OsOFP* genes, the phenotypes of their mutants are related to hormonal pathways. The *OsOFP* genes regulate plant structure and grain shape by influencing hormone signaling or biosynthesis pathways, particularly the gibberellin (GA) and brassinolide (BR) pathways. However, their specific roles in regulating these pathways differ. For example, *OsOFP1* and *OsOFP8* act as positive regulators of the BR response. Plants overexpressing *OsOFP1* and *OsOFP8* exhibit reduced plant height and increased lamina inclination in rice seedlings. They are also more sensitive to BR treatment [5,6]. OsOFP1 interacts with the DLT (DWARF and LOW-TILLERING) transcription factor and OsGSK2 protein kinase to participate in the BR signaling pathway [6]. Meanwhile, the application of exogenous GA to homozygous transgenic rice seedlings overexpressing *OsOFP1* can restore the phenotype of *OsOFP1*-overexpressing plants. This suggests that the *OsOFP1* gene may inhibit the normal growth and development of plants by inhibiting GA synthesis [11]. OsOFP8 plays a positive role in the BR signaling pathway by interacting with OsGKS2, a negative regulator of the BR signaling pathway [5]. By contrast, OsOFP3, OsOFP19, and OsOFP22 are negative regulators of the BR response. The stability of the OsOFP3 protein is regulated by BR and GSK2 [8]. Plants overexpressing *OsOFP19* are insensitive to exogenous BR treatment. This gene negatively regulates the BR response through interactions with DLT and OSH1 (Oryza sativa homeobox1) and integrates it with cell division patterns to influence plant structure and grain shape [7]. Plants overexpressing *OsOFP22* are insensitive to both exogenous BR and GA. *OsOFP22* regulates the structure of rice plants and the shape of grains by inhibiting GA and BR signal transduction [9]. Overexpressing *OsOFP2* under the control of the 35S promoter reduces plant height and alters leaf morphology, grain shape, and the positioning of vascular bundles in the stem [10]. Sequence analysis revealed cis-acting elements associated with abscisic acid (ABA) and GA synthesis in the promoter region of *OsOFP2*. *OsOFP2* overexpression inhibits GA biosynthesis and OsOFP2 interacts with KNOX and BELL proteins to regulate vascular bundle development [10].

The *OFP* genes of other plants have similar functions to those of *Arabidopsis* and rice. For example, MaOFP1 interacts with MuMADS1 to regulate banana fruit development [12]. *SlOFP20* can respond to exogenous GA and BR, and overexpression of *SlOFP20* causes GA insensitivity and the phenotype of BR deletion [13,14].

*OsOFP6* (*LOC_Os01g60810*) is an *OFP* gene in rice. In a previous study, Ma et al. constructed an *OsOFP6* RNAi vector and a 35S-driven overexpression vector in the context of wild-type rice ZH11 [12]. *OsOFP6* RNAi resulted in a semi-dwarf stature with an altered grain shape and shortened lateral roots. By contrast, overexpressing plants were not significantly different from the wild-type [12,15]. Here, we isolated a rice mutant that displayed short and wide grains, erect leaves, and reduced plant height. Gene cloning and functional verification confirmed that the mutation was caused by the overexpression of *OsOFP6*. Transgenic plants overexpressing *OsOFP6* had a similar phenotype to those overexpressing *OsOFP3*, *OsOFP19*, and *OsOFP22*, while *OsOFP6* knockout plants were not significantly different from the wild-type, which were inconsistent with previous findings about *OsOFP6* and therefore prompted us to further investigate this finding.

In this study, we combined phenotypic analysis, hormone response, real-time PCR, subcellular localization, and protein interaction experiments to investigate the function of *OsOFP6*. We aim to understand how *OsOFP6* affects rice growth and development and its underlying molecular mechanism.

## 2. Results

### 2.1. T-DNA Insertion Co-Segregates with the Osofp6-D Mutant

By screening a library of rice T-DNA insertion mutants, we identified a transgenic line that carries a semi-dominant dwarf mutation. By amplifying the hygromycin phosphotransferase gene, we discovered that the T_1_ and T_2_ generation mutants of the transgenic line contained T-DNA, whereas the wild-type did not (Appendix A). Therefore, we speculated that this mutation was caused by T-DNA insertion.

To identify the flanking sequence of the T-DNA insertion, TAIL-PCR analysis was performed. Through sequence analysis of the TAIL-PCR products, we discovered that the T-DNA was inserted approximately 4.4 kb downstream of the *OsOFP6* gene (*LOC_Os01g60810*) (Figure 1A), which encodes a protein belonging to the OVATE family. The T-DNA insertion caused an increase in *OsOFP6* expression (Figure 1B). Thus, the mutant was named *Osofp6-D*.

Sixty F_2_ generation plants, including 20 wild-type, heterozygous, and homozygous mutants each, were genotyped to further determine whether the *Osofp6-D* phenotype co-segregated with the T-DNA insertion. Homozygous mutants were identified as homozygous for the T-DNA insertion because only the 390-bp bands were amplified by the P1 and P2 primer pairs (lanes A1–A8). Both 390- and 441-bp bands were amplified from heterozygous mutants, indicating the presence of T-DNA insertion heterozygosity (lanes B1–B8). Wild-type plants did not have T-DNA insertion because only 441-bp bands were amplified (lanes C1–C8) (Figure 1C,D). These results further suggest that the *Osofp6-D* phenotype is caused by T-DNA insertion.

### 2.2. Overexpression of OsOFP6 Alters Plant Architecture and Grain Shape

The T_1_ progeny of the transgenic line containing *Osofp6-D* segregated into wild-type, weakly dwarfed, and severely dwarfed plants. This indicates that the *Osofp6-D* mutant is a heterozygous semi-dominant mutation. The homozygous mutant was further crossed with its wild-type parent, Nipponbare, and all plants in the F_1_ generation exhibited the mutant phenotype. In the F_2_ generation, the segregation ratio of wild-type, weakly dwarfed, and severely dwarfed plants showed a good fit to a 1:2:1 ratio (normal: weak: severe = 34:68:26, X^2^ = 1.5 < X^2^0.05 = 5.99). The results further showed that the *Osofp6-D* mutation was controlled by a single semi-dominant gene.

The phenotype of the mutant was further characterized by F_2_ generation plants under natural field conditions. Compared to the wild-type, both the heterozygous and homozygous mutant plants showed significant morphological changes at the mature stage (Figure 2A). The average plant height was reduced to 81.7% and 70.1% of the wild-type for the heterozygous and homozygous mutants, respectively (Figure 2B); the tiller number was reduced to 83.4% and 39.4% of the wild-type, respectively, but there was no significant difference between the heterozygous mutant and wild-type (Figure 2C); The main panicle length was reduced to 82.7% and 74.3% of the wild-type, respectively (Figure 2D,E); and the leaf angle was reduced to 48.0% and 47.1% of the wild-type, respectively (Figure 2F,G). There were significant differences in the plant height, tiller number, and main panicle length between the homozygous and heterozygous mutants, but there was no significant difference in the leaf angle (Figure 2B,C,E,G).

Compared to the wild-type, both the heterozygous and homozygous mutant plants showed decreased grain length and increased grain width (Figure 3A–C). The heterozygous mutant had seeds that were 9.4% wider and 16.2% shorter than those of the wild-type (Figure 3D,E). However, there was no significant difference in 1000-grain weight between the mutant and wild-type (Figure 3F). To investigate the cytological causes of the altered seed shape in the *Osofp6-D* mutant, SEM was used to analyze the outer epidermal cells of rice glumes. The mutant cell length (the distance between the two bulges) was reduced, and the cell width was significantly increased (Figure 3G–I). This suggests that *OsOFP6* overexpression alters the grain shape by affecting the shape of the seeds and thus, the grain shape. These results indicate that the overexpression of *OsOFP6* alters plant architecture and grain shape.

### 2.3. Overexpression of OsOFP6 Affects the Seed Setting Rate and Seed Germination

To explore the reasons for the decreased seed setting rate in the *Osofp6-D* mutant, the pollen viability of the wild-type and *Osofp6-D* mutant at the young spike stage was tested. The pollen number and stainability of the mutant were significantly lower compared to the wild-type (Figure 4A,B). The germination times of the wild-type and the mutant were measured. The germination time of the *Osofp6-D* mutant was about 24 h later than that of the wild-type, and it also had a significantly lower germination rate than that of the wild-type (Figure 4C,D). These results indicate that *OsOFP6* overexpression affects the seed setting rate and germination.

### 2.4. Functional Verification of OsOFP6

To validate that *OsOFP6* overexpression leads to phenotypic changes in the *Osofp6-D* mutant, we obtained the *Osofp6-D* knockout plants (*Osofp6-D-kn1* and *Osofp6-D-kn2*) from the *Osofp6-D* mutants using the CRISPR/Cas9 system. The phenotype of the *Osofp6-D-kn1* and *Osofp6-D-kn2* plants was not significantly different from that of the wild-type (Figure 5A,B), suggesting that the phenotype of the mutant was due to *OsOFP6* overexpression.

To further confirm the function of *OsOFP6*, a 35S promoter-driven *OsOFP6* overexpression vector was constructed and transformed into wild-type Nipponbare to obtain *OsOFP6*-overexpressing plants. Plants overexpressing *OsOFP6* with the 35S enhancer mimicked the *Osofp6-D* phenotypes (Figure 6A–C). The height and main panicle length of the overexpressing plants decreased to 49.8% and 47.0% of that of the wild-type, respectively, showing a semi-dwarfed phenotype (Figure 6D,E). In addition, the tiller number and leaf angle of *OsOFP6-OE* were also significantly altered; the tiller numbers were 12 and 4, respectively, reduced to 33.3% of that of the wild-type (Figure 6F). The wild-type and overexpressing plants had leaf angles of 23.5° and 10.1°, respectively, showing a reduction to 43.0% of the wild-type (Figure 6G). These results indicate that *OsOFP6* overexpression can reduce plant height, main panicle length, tiller number, and leaf angle.

Phylogenetic analysis indicated that OsOFP6 was most closely related to OsOFP22 in rice [1]. Plants overexpressing *OsOFP6* showed a similar phenotype to those overexpressing *OsOFP22* (Figure 2) [9]. To elucidate the role of the *OsOFP6* gene, a CRISPR/Cas9 vector targeting *OsOFP6* and *OsOFP22* simultaneously was constructed [16] and transformed into the wild-type using the *Agrobacterium tumefaciens*-mediated genetic transformation method. Eight homozygotes with frameshift mutations in both *OsOFP6* and *OsOFP22* genes were identified by sequencing. However, none of these plants showed obvious changes in plant architecture and grain shape compared to those of the wild-type plants.

### 2.5. Expression Pattern of OsOFP6 and Subcellular Localization of OsOFP6

To investigate the expression pattern of *OsOFP6*, various tissues were sampled from wild-type plants for qRT-PCR analysis. *OsOFP6* was expressed in all tested tissues. The highest expression level of *OsOFP6* was found in leaf blades, followed by internode, root, panicle, and leaf sheath (Figure 7A). To determine the subcellular localization of the *OsOFP6* gene-encoding protein, a 35S:OsOFP6-GFP fusion expression vector carrying green fluorescent protein was constructed by recombining the coding sequence of *OsOFP6* with a 35S:GFP expression vector. The fusion expression vector was transformed into Agrobacterium GV3101 and injected into *Nicotiana benthamiana* leaves for transient expression. Nuclear localization markers and membrane localization markers were labeled and observed under a Zeiss LSM700 confocal microscope. As shown in Figure 7B, green fluorescence was distributed in whole cells in *Nicotiana benthamiana* leaves transformed with the control vector 35S:GFP. OsOFP6 fluorescence was co-located with the nuclear localization marker and membrane localization marker, indicating that the *OsOFP6* gene-encoded protein was located in the nucleus and cell membrane.

### 2.6. Plants Overexpressing OsOFP6 Display Less Sensitivity to GA

The phenotype of the *Osofp6-D* mutant was similar to that of the GA- or BR-defective mutants (Figure 2). To determine whether the *Osofp6-D* mutant responded to exogenous GA, wild-type Nipponbare and the *Osofp6-D* mutant were treated with different concentrations of GA3, and changes in the length of the second leaf sheath and plant height of the seedlings were observed 7 days after germination. As shown in Figure 8, the length of the second leaf sheath and plant height of *Osofp6-D* were always significantly shorter than the wild-type under different exogenous GA3 concentrations (Figure 8A,B). The results indicate that plants overexpressing *OsOFP6* are less sensitive to exogenous GA.

In addition, we performed a GA3-induced α-amylase assay. The starch in the medium was hydrolyzed by α-amylase. After staining with I_2_-KI, a clear halo formed around the endosperm. The larger the diameter of the clear halo, the more α-amylase was produced. The slight variation in the size of the clear halo in the same plate was due to the difference in vigor of the same batch of seeds. The results showed that α-amylase was not induced in the wild-type or *Osofp6-D* mutants on the plate without GA3. On the starch plate with exogenous 10^−6^ M GA3, the clear halo diameter of the *Osofp6-D* mutant hydrolyzing α-amylase was significantly reduced compared to that of the wild-type, indicating that the *Osofp6-D* mutant produced less α-amylase than that of wild-type Nipponbare (Figure 8C). Plants overexpressing *OsOFP6* were less sensitive to GA treatment, and *OsOFP6* overexpression might have inhibited the GA reaction.

To clarify whether the rice *Osofp6-D* phenotype is related to the GA pathway, qRT-PCR was performed to detect the expression of genes related to GA synthesis and signaling pathways in the wild-type and *Osofp6-D* mutants. The expression of the GA biosynthesis-related genes *OsCPS*, *GA3ox1*, *GA13ox*, and *GA20ox3* significantly increased in *Osofp6-D*, while the expression of *OsGID2*, a GA signal transduction-related gene, decreased (Figure 8D). Previous studies have shown that GA signaling negatively regulates the expression of GA biosynthesis-related genes. When the GA signaling pathway is blocked, GA biosynthesis-related gene expression is elevated to accumulate more GA-active material. In summary, *OsOFP6* acts as a negative player in GA signaling to inhibit the GA response.

### 2.7. Plants Overexpressing OsOFP6 Display Less Sensitivity to BR

In previous reports, the BR-deficient phenotype showed reduced plant height, leaf angle, and compact plant architecture. The *Osofp6-D* mutant also showed decreased plant height and leaf angle (Figure 2). We speculated that *OsOFP6* might be involved in BR pathways. We measured the length of the roots under different concentrations of 24-epiBL (24-epibrassinolide). The results showed that *Osofp6-D* plant roots had decreased BR sensitivity (Figure 9A,B). To further confirm that *Osofp6-D* is a less sensitive BR mutant, lamina inclination assays were performed. While the leaf angles of the wild-type plants increased following BR application, those of the *Osofp6-D* lines were poorly responsive to BR treatment (Figure 9C), suggesting that the BR response was inhibited in *Osofp6-D* plants. Combined with the typical BR-deficient phenotypes of plants, these results suggest that *OsOFP6* overexpression inhibits BR signaling.

To further confirm the negative role of *OsOFP6* in the BR pathway, we analyzed the expression levels of the BR biosynthesis gene and the BR signal transduction gene in the wild-type and *Osofp6-D* plants. Compared to the wild-type, the BR signal transduction gene *BZR1* significantly decreased in the mutant, and the BR biosynthesis genes *D2*, *OsCPD*, and *DWARF* significantly increased, which was consistent with the reduced BR sensitivity phenotype of the mutants (Figure 9D). These results demonstrated that *OsOFP6* overexpression inhibits the BR response and BR signaling gene expression.

### 2.8. OsOFP6 Physically Interacts with E3BB

To explore the molecular pathway of OsOFP6 involvement, we performed a yeast two-hybrid screening assay to identify interacting partners. We found that the *LOC_Os09g35690* gene encodes an E3 ubiquitin-protein ligase, BIG BROTHER, which we named *E3BB*. Its function in rice is unknown, but in *Arabidopsis*, the *BB* gene represents a central regulator of organ size. Plants lacking *BB* activity form larger organs. Conversely, plants that express higher levels of *BB* produce smaller organs, suggesting that *BB* is a negative regulator of organ size [17].

The yeast two-hybrid assays confirmed the interaction of the full-length OsOFP6 with E3BB (Figure 10A). We performed a LUC complementation imaging assay. A distinct LUC activity signal was generated When cLUC-OsOFP6 and E3BB-nLUC were co-expressed in the leaf epidermal cells of *Nicotiana benthamiana* (Figure 10B). Using the bimolecular fluorescence complementation (BiFC) assay, we observed a strong yellow fluorescent protein signal when OsOFP6-SPYNE and E3BB-SPYCE were co-expressed within the leaf epidermal cells of *Nicotiana benthamiana*. (Figure 10C). The results showed that OsOFP6 interacted with E3BB in plant cells.

## 3. Discussion

OVATE family proteins are plant-specific regulatory proteins involved in regulating plant growth and development [2]. *OsOFP6* belongs to the rice OVATE family gene. Previous studies have shown that RNA interference (RNAi) of *OsOFP6* led to large leaf angles, slender seeds, and short lateral roots. No significant phenotypic changes were observed between *OsOFP6*-overexpressing and wild-type plants. This study identified a novel semi-dominant mutant, *Osofp6-D*, which overexpressed *OsOFP6* and was derived from a T-DNA insertion library. The *Osofp6-D* mutant had several unique agronomic traits, including short stalks, erect leaves, short and wide grain shape, reduced fertility, and delayed seed germination. However, no significant phenotypic changes existed between the knockout plants of *OsOFP6* and the wild-type. The *Osofp6-D* mutant showed reduced sensitivity to GA and BR and inhibited the GA and BR signaling pathways (Figure 8 and Figure 9). OsOFP6 can also interact with E3BB (Figure 10).

Ma et al. showed that, in the ZH11 background, *OsOFP6* gene expression was inhibited by RNA interference (RNAi), resulting in reduced plant height, narrower seeds, and shorter lateral roots [12]. There was no significant difference between 35S-overexpressing plants and wild-type ZH11 [12]. In this study, the knockout mutant of *OsOFP6* obtained by the CRISPR/Cas9 system showed no significant differences in phenotype compared to the wild-type, but the phenotype of *OsOFP6*-overexpressing plants was significantly different from that of the wild-type, and the phenotype of transgenic plants overexpressing *OsOFP6* with the 35S promoter was consistent with that of *Osofp6-D* (Figure 5 and Figure 6). It was speculated that Ma et al. obtained *OsOFP6* RNAi plant phenotypes through RNAi, which might interfere with *OsOFP6* and multiple homologous genes, thus producing a variety of phenotypes [12]. Although both of the overexpressing plants were activated by the 35S promoter, the background materials of experimental studies were inconsistent, suggesting that there might be a potentially unknown complex compensatory pathway in the cell, leading to differences in overexpressing plant types. In this study, *OsOFP6* knockout using the CRISPR/Cas9 system was used to restore the phenotype of the *Osofp6-D* mutant, which confirmed that *OsOFP6* overexpression led to dwarfing, shortening, and widening of the grain and other polymorphic phenotypes (Figure 5).

Liu et al. performed a phylogenetic analysis and found that OsOFP6 was highly homologous to OsOFP22 and OsOFP19 by comparing the conserved OVATE domain [1,18]. Overexpression of *OsOFP22* and *OsOFP19* resulted in plant dwarfism, reduced main spike length, reduced leaf angle, and short and wide grains [7,9]. There was no significant difference between *OsOFP19* RNAi plants and the wild-type (ZH11) [7]. In this study, plants overexpressing *OsOFP6* altered plant architecture and grain shape similar to those overexpressing *OsOFP22* and *OsOFP19* (Figure 2 and Figure 3). The *Osofp6Osofp22* knockout plants obtained using the CRISPR/Cas9 system showed no significant differences from the wild-type (Nipponare). It suggest that *OsOFP6*, *OsOFP22,* and *OsOFP19* are functionally redundant in regulating plant architecture and grain shape in rice. In addition, we found that overexpression of *OsOFP6* affects pollen fertility and seed setting rate (Figure 4), but it is not known whether overexpression of *OsOFP22* and *OsOFP19* has similar phenotypes.

In previous studies, a feedback mechanism between GA and BR has been shown to regulate plant growth in rice [19]. GA and BR crosstalk in *Arabidopsis* through direct interactions between GA-inactivated DELLAs and BR-activated BZR1 [20,21]. *OsOFP22*-overexpressing plants were semi-dwarfed with dark green leaves and short and wide flowering organs and grains and had reduced sensitivity to GA and BR. Further studies showed that *OsOFP22* overexpression promoted protein accumulation of the rice DELLA gene *SLR1*, which may interact with BZR1 to suppress its transcriptional activity, thereby inhibiting the BR response and ultimately regulating rice plant architecture and seed shape by affecting crosstalk between GA and BR [9]. Therefore, we performed the GA-induced α-amylase assay and second leaf sheath elongation assay, as well as the BR-induced leaf inclination assay and the root waving assay on wild-type Nipponare and the *Osofp6-D* mutant. Plants overexpressing *OsOFP6* were less sensitive to GA and BR, consistent with *OsOFP22* (Figure 8 and Figure 9). The biosynthesis of GA and BR is regulated by the negative feedback of signal transduction. The transcription level of the synthetase gene is upregulated when the hormone level in tissues is reduced or when signal transduction is blocked. When the hormone level in cells is increased or signal transduction is enhanced, the expression level of the synthetase gene is downregulated, and GA can inhibit BR biosynthesis and BR responses [15,19,22,23]. As shown in Figure 8E and Figure 9E, the expression levels of GA and BR biosynthesis-related genes and the downregulation of signal transduction genes in the wild-type and *Osofp6-D* mutants also suggest defects in the OsOFP6 signaling pathway, which may inhibit GA and BR signal transduction responses. The expression level of the GA key gene *SLR1* was significantly upregulated, and the expression level of the BR signaling gene *BZR1* was significantly downregulated. It is speculated that OsOFP6 may also affect the SLR1-BZR1 pathway and the regulation of plant architecture and grain shape through GA and BR crosstalk.

OsOFP8 and OsOFP19 are the positive and negative factors of the BR signaling pathway, respectively. The F-box protein FBX206 interacts with OsOFP8 and OsOFP19, respectively, and oppositely regulates their stability [24]. OsOFP19, OSH1, and DLT interact with each other to form a functional complex and are involved in the BR signaling pathway [7]. OsGSK2, a conserved GSK3-like kinase serving as a key suppressor in BR signaling, interacts with and phosphorylates OsOFP8 [5]. These genes form a complex network to regulate plant growth and development by BR signaling. OsOFP6 and OsOFP19 are homologous proteins with highly similar sequences and structures. Plants that overexpress *OsOFP6* and *OsOFP19* show a similar phenotype (Figure 2 and Figure 3). Therefore, we speculate that OsOFP6 may be also a member of this network and participate in the BR signaling pathway.

Research has shown that BR promotes seed germination and post-germination growth. The germination of rice seeds of the *d61*, *Go*, and *bzr1* BR-insensitive signal mutants or transgenic rice lines was delayed, and the length of the shoot was shorter than that of the wild-type [25,26]. BR promotes germination in rice seeds via the key transcription factor BZR1, which mediates the expression of downstream target genes. One representative direct target of BZR1 is *RAmy3D*, a gene that encodes an important α-amylase that plays a crucial role in starch degradation in the initial stage of seed germination [26]. In this study, *Osofp6-D* showed low sensitivity to exogenous BR, and the germination time of *Osofp6-D* was delayed by about 24 h. Compared to the wild-type, the expression of *BZR1*, a key gene for BR signal transduction, was downregulated in the mutant, indicating that *Osofp6-D* was involved in the BR signal pathway. It is possible to delay the germination time of the *Osofp6-D* mutant by affecting the expression of BZR1.

We identified an OsOFP6-interacting protein, E3BB, the rice ortholog of the *Arabidopsis* E3 ubiquitin protein ligase BIG BROTHER (BB). Its function in rice is unknown, but in *Arabidopsis thaliana*, the *BB* gene represents a central regulator of organ size. *BB* expression mirrors proliferative activity, yet the gene functions to limit proliferation, suggesting that it acts in an incoherent feedforward loop downstream of growth activators to prevent over-proliferation. Subsequent studies have shown that the expression of the central growth regulator *BB* is regulated by multiple cis-elements [27]. Plants that lack BB activity form larger organs. Conversely, plants that express high levels of *BB* produce smaller organs, suggesting that BB is a negative regulator of organ size [17]. The BB protein has E3 ubiquitin-ligase activity and can activate DA1, DAR1, and DAR2 through mono-ubiquitination at multiple sites. Subsequently, these activated peptidases destabilize various positive growth regulators and limit the duration of organ growth and ultimately, organ size [28]. Therefore, we speculate that the interaction of OsOFP6 and E3BB may activate the activity of E3BB and negatively regulate rice growth and development, but the specific regulatory pathway remains to be investigated. The specific functions of these two genes in rice will be studied in the future.

## 4. Materials and Methods

### 4.1. Plant Materials and Phenotypic Analysis

The *Osofp6-D* mutant was isolated from a rice T-DNA insertion mutant library containing 15,000 transgenic rice (*Oryza sativa* L. ssp. *japonica* cv. Nipponbare) lines [29]. Rice plants were grown in a paddy field at the China National Rice Research Institute (119°57′ E, 30°03′ N). The mutant was used as the female parent, and the wild-type Nipponbare was used as the male parent for hybridization. The above experimental materials received conventional water and fertilizer management. F1 and F2 generation plants from a cross of the mutant and wild-type were used for genetic and phenotypic analysis. The plant height, panicle length, tiller number, grain length, grain weight, seed setting rate, and 1000-seed weight were investigated.

### 4.2. TAIL-PCR

Adjacent sequences were obtained by thermal asymmetric interleaved PCR (TAIL-PCR) [30]. The specific primer TL was used to isolate the left boundary sequence of T-DNA, and the primer sequence is shown in Appendix A. Three rounds of PCR amplification were performed. The products obtained in the third round were cloned into a pMD18-T vector, and the samples were sent to the Beijing Tsingke Biotech Co., Ltd. (Beijing, China) for sequencing.

### 4.3. Vector Construction and Rice Transformation

To construct the complementary vector, a CRISPR/Cas9 vector targeting the *OsOFP6* gene was constructed. A 20-bp fragment of the *OsOFP6* gene was assembled into the intermediate vector SK-gRNA. The gRNA was then inserted into the CRISPR/Cas9 binary vector pCAMBIA2300-Cas9 [16]. The primers are listed in Appendix A.

To construct an overexpression vector, the full-length coding sequences of *OsOFP6* were amplified by PCR and inserted into the pCAMBIA130035S-3×Flag binary vector. The two constructs were introduced into rice using the *Agrobacterium tumefaciens*-mediated genetic transformation method [31]. The primers are listed in Appendix A.

### 4.4. DNA Extraction and PCR

The total DNA of rice was extracted from the fresh leaves of plants using the CTAB method [32]. For mutant identification, two primers were designed to amplify the hygromycin phosphotransferase gene and to identify whether it was a T-DNA insertion mutant. Three primers were designed to determine whether the *Osofp6-D* phenotype co-segregated with the T-DNA insertion. The PCR reaction was carried out as follows: denaturation at 95 °C for 3 min, followed by 30 cycles of 95 °C for 15 s, annealing at 56 °C for 30 s and 72 °C for 35 s, and a final extension step at 72 °C for 7 min. The primers are listed in Appendix A.

### 4.5. RNA Extraction and Quantitative Real-Time PCR (qRT-PCR)

Total RNAs were isolated from various plant tissues using an RNAprep Pure Plant Kit (TIANGEN, Beijing, China). First-strand cDNA was synthesized from 1 μg of total RNA using the Transcriptor cDNA Synth. Kit 2 (Roche, Mannheim, Germany). Quantitative RT-PCR assays were performed on a Bio-Rad CFX96TM real-time system using a FastStart Universal SYBR Green Master (Roche, Mannheim, Germany), and the *ACTIN1* gene was used as an internal control. The relative expression was calculated using 2^−ΔΔCT^. The primers for the qRT-PCR assay are listed in Appendix A.

### 4.6. Phenotypic Evaluation and Cellular Analysis

The plants of Nipponbare and *Osofp6-D* mutants at maturity were photographed. The grain length, width, and 1000-grain weight were measured using a rice appearance quality detector (WSeen, Hangzhou, China) and repeated three times. Scanning electron microscopy (SEM) was used to observe the cell size. The mature rice seed glumes were cleaned of surface dust and tightly attached to a conductive carbon film double-sided adhesive before being placed on the ion sputterer sample stage for 30 s. They were then observed via SEM S-3400N (Hitachi, Tokyo, Japan) [33]. Cell length and width were measured using ImageJ 1.8.0 software.

### 4.7. Pollen Vitality Staining Experiments

For the pollen vitality staining experiment, florets at the young spike stage of rice were collected. The glume was removed, and the anthers were crushed with tweezers, placed on a slide with 1% I_2_-KI solution, and observed under an optical microscope [34].

### 4.8. Subcellular Localization

To determine the subcellular localization of the OsOFP6 protein, the full-length coding sequences of *OsOFP6* (removal of termination codons) were fused into the N-terminus of the coding region of the green fluorescent protein (GFP) in the GFP vector and were driven by the CaMV 35S promoter to generate the 35S:OsOFP6-GFP vector. The construct was introduced into *Agrobacterium* strain GV3101 and used to infiltrate the leaves of *Nicotiana benthamiana* plants [35]. GFP fluorescence was observed using a Zeiss LSM700 laser scanning confocal microscope (Zeiss, Thornwood, NY, USA). The primers are listed in Appendix A.

### 4.9. Seed Germination Trials

Twenty full rice seeds were selected and placed in a 7-cm diameter glass dish lined with filter paper. An appropriate amount of water was added, and the plates were incubated in a light incubator at 30 °C. Seed germination was observed on the second day of incubation, and germination was recorded at 24-h intervals. Each material was repeated three times.
Germination rate = (number of germinated seeds/total number of seeds) × 100%

### 4.10. GA3 Treatment

For α-amylase activity assessment, an agar plate assay with α-amylase was performed, as described previously [36]. The rice seeds were hulled with a hulling machine, and the half that contained the embryo was removed using a blade to cut across the seeds. The half seeds without embryos were disinfected with 5% NaClO on an ultra-clean table for 30 min. The waste liquid was poured away, and the half seeds were washed with sterile water 2–3 times for 10 min each time to wash away the remaining NaClO. The sterilized filter paper was used to absorb the moisture on the surface of the seeds, and the half-seeds were placed in a 2% starch agar medium without GA3 or with 10^−6^ M GA3 (G500, PhytoTech LABS, Lenexa, KS, USA) with sterilized tweezers, with the seed section close to the surface of the medium, and then incubated in an incubator at 30 °C for 48 h. After 48 h, the seeds were extracted, and the starch agar plates were stained with 0.1% I2 and 1% KI solution to detect the activity of α-amylase released from the seeds. The clear halo around the endosperm was the result of starch hydrolysis by α-amylase synthesized and secreted by the endosperm.

For the second leaf sheath elongation assay, which was slightly modified from that described previously [37], the sterilized Nipponbare and *Osofp6-D* mutants were grown in Yoshida’s culture solution (NSP1040, Coolaber, Beijing, China) containing 0, 10^−6^, 10^−5^, and 10^−4^ M GA3 and incubated at 30 °C with 12 h of light/ 12 h of darkness. After one week of growth, the length of the second leaf sheath was measured.

### 4.11. BR Treatment

For the lamina joint bending assay, as described previously [6], full, mature, mold-free rice seeds were selected and processed through the following steps: the seeds were disinfected twice with 75% alcohol, washed three times with sterile water, and then placed in a constant-temperature incubator at 30 °C for 2 days, changing the water once a day. After germination, seeds of uniform growth were selected and placed in a hydroponic box containing a rice nutrient solution. After 7 d of incubation, uniformly growing seedlings were sampled and cut to contain approximately 2.5 cm segments of the second leaf joint, leaf blade, and leaf sheath. They were then placed in sterilized Petri dishes with 15 mL of various concentrations of 24-epiBL (Sigma-Aldrich, Shanghai, China) solution (0, 10^−6^, 10^−5^, and 10^−4^ M), and the Petri dishes containing the leaves were placed in an incubator at 30 °C and incubated in the dark for 48 h. The leaf angles were measured using ImageJ.

The BR-induced root-waving assay was performed as described previously [38], with slight modifications. Full, mature, mold-free rice seeds were selected and manipulated as follows: after disinfection with 75% alcohol twice and rinsing with sterile water three times, the seeds were placed onto 1/2MS medium with forceps, germinated in a constant-temperature incubator at 30 °C for 2 d, and grown for 3 d. Uniformly germinated seeds were selected and grown in 1/2MS without 24-epiBL and 1/2MS medium containing 10^−6^ M 24-epiBL, respectively. The seeds were grown in 1/2MS medium without 24-epiBL for 7 d. The phenotypes were observed, and root lengths were measured afterward.

### 4.12. Protein Interaction Experiments

For yeast two-hybrid assays, the full-length coding sequences of *OsOFP6* were cloned into pGBKT7 DNA-BD as the bait, and the full-length coding sequences of *E3BB* were cloned into pGADT7 as the prey. The prey and bait constructs were co-transformed into Y2HGold yeast cells. Then, cells transformed with AD-E3BB and BD-OsOFP6 constructs were selected by growing them on a selective medium (−Leu/−Trp). The growth of the selected cells was examined on a selective medium (−Leu/−Trp/−His-Ade) to confirm the interaction between OsOFP6 and E3BB. The transformation was performed using the Matchmaker Gold Yeast Two-Hybrid System (Clontech, Mountain View, CA, USA) according to the manufacturer’s instructions.

For bimolecular fluorescence complementation (BiFC) assays, the full-length coding sequences of *OsOFP6* (removal of termination codons) and *E3BB* (removal of termination codons) fused with the N-terminal and C-terminal fragments of *YFP*. The OsOFP6-SPYNE and E3BB-SPYCE constructs and empty vector controls were transformed into *Agrobacterium* strain GV3101. Equal amounts of bacteria carrying the SPYCE or SPYNE fusion plasmid were combined and infiltrated into young *Nicotiana benthamiana* leaves, as previously described [39]. After 48 h, confocal imaging analysis was performed using an LSM 700 confocal microscope (Zeiss, Thornwood, NY, USA).

For LUC complementation imaging (LCI) assays, the full-length coding sequences of *OsOFP6* and *E3BB* (removal of termination codons) were fused with the C-terminal and N-terminal fragment of the luciferase reporter gene. The cLUC-OsOFP6 and E3BB-nLUC constructs and empty vector controls were transformed into *Agrobacterium* strain GV3101. Equal amounts of bacteria carrying the cLUC or nLUC fusion plasmid were combined and infiltrated into young *Nicotiana benthamiana* leaves. After 48 h, the abaxial surface of the leaf was sprayed with a reaction solution of 1 mmol∙L^−1^ D-luciferin and then left in the dark for 7 min. The LUC fluorescent signals in the infiltrated leaves were analyzed by chemiluminescence imaging (Tanon 5200, Shanghai, China). The primers are listed in Appendix A.

### 4.13. Statistical Analysis

All numerical data are presented as the means ± standard deviations (SDs, represented by error bars). Statistical analyses were carried out using Excel 2016 (Microsoft, Redmond, WA, USA) and GraphPad 9 software. The differences between transgenic and wild-type plants were determined using Student’s *t*-test (* *p* < 0.05; ** *p* < 0.01; ns: not significant).

## 5. Conclusions

Our results suggest that overexpression of *OsOFP6* regulates plant architecture, grain shape, and seed fertility. Plants that overexpress *OsOFP6* exhibit reduced sensitivity to GA and BR treatments. OsOFP6 may redundantly regulate plant architecture and grain shape with its homologs. OsOFP6 is located in both the nucleus and cell membrane and interacts with the E3 ubiquitin ligase E3BB, but it is unclear how their interaction affects phenotype. This study further elucidates the regulatory mechanism of the rice OFPs on plant architecture and grain shape. It provides important reference values for establishing plant types and improving grain shape.

## Figures and Tables

**Figure 1 ijms-25-02889-f001:**
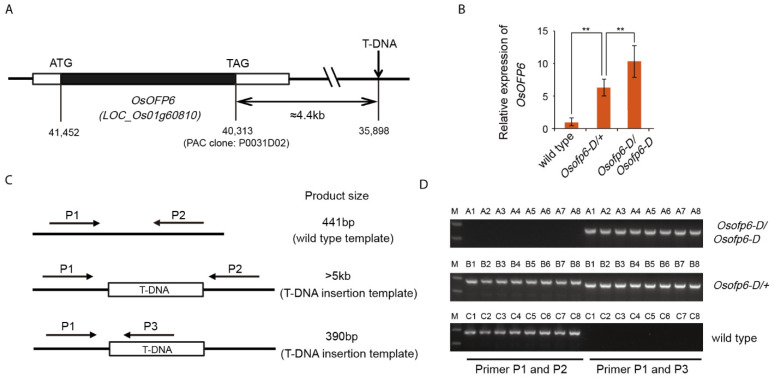
Gene structure of *Osofp6-D* and co-segregation analysis of T-DNA and *Osofp6-D*. (**A**) Schematic representation of *Osofp6-D*. The T-DNA was inserted approximately 4.4 kb downstream of the *OsOFP6* gene, which contains one exon and encodes a 380-amino acid protein. (**B**) Expression analysis of *OsOFP6* in wild-type, heterozygous mutants (*Osofp6-D/+*) and homozygous mutants (*Osofp6-D/Osofp6-D*). Error bars indicate means ± SD, *n* = 3 (** *p* < 0.01, Student′s *t*-test) (**C**) Schematic diagrams of genotyping. The P1 and P2 primers amplified a 441 bp fragment from the wild-type DNA template, but they did not amplify any PCR band from the T-DNA insertion template because the expected fragment containing the T-DNA was too large to amplify; P1 and P3 primers amplified a 390-bp PCR band from the T-DNA insertion DNA template. (**D**) Genotyping of F2 plants. A1–A8: homozygous mutants (*Osofp6-D*/*Osofp6-D*); B1–B8: heterozygous mutants (*Osofp6-D*/+); C1–C8: wild-type plants; M: marker.

**Figure 2 ijms-25-02889-f002:**
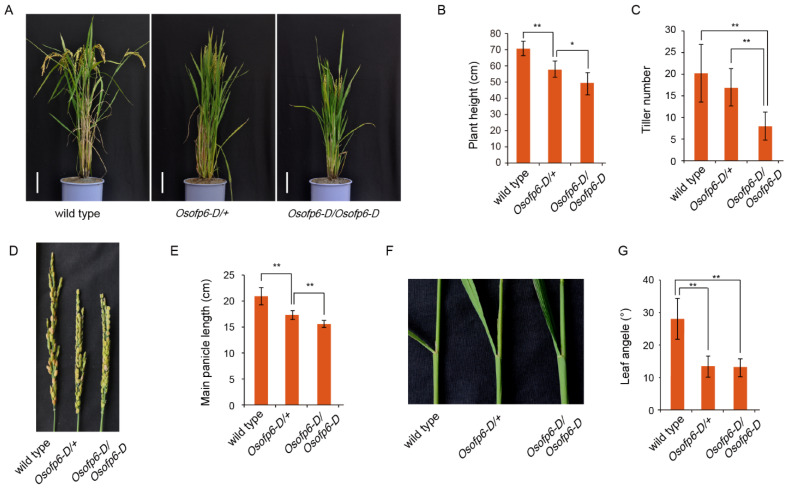
Overexpression of *OsOFP6* affects plant morphology. (**A**) Morphologies of the wild-type (Nipponbare), heterozygous mutants (*Osofp6-D*/+), and homozygous mutants (*Osofp6-D*/*Osofp6-D*) at the mature stage. Scale bar: 10 cm. (**B**) Statistical analysis of plant height. (**C**) Statistical analysis of the tiller number. (**D**) Morphologies of the main panicle at the mature stage of the wild-type, *Osofp6-D*/+, and *Osofp6-D*/*Osofp6-D*. (**E**) Statistical analysis of the main panicle length. (**F**) Morphology of leaf angle in the wild-type, *Osofp6-D*/+, and *Osofp6-D*/*Osofp6-D*. (**G**) Statistical analysis of the leaf angle. Error bars indicate the means ± SD, *n* = 15. Statistical differences between the wild-type and *Osofp6-D* mutants (*Osofp6-D*/+, *Osofp6-D*/*Osofp6-D*) are indicated by asterisks and were determined by Student’s *t*-test (* *p* < 0.05; ** *p* < 0.01).

**Figure 3 ijms-25-02889-f003:**
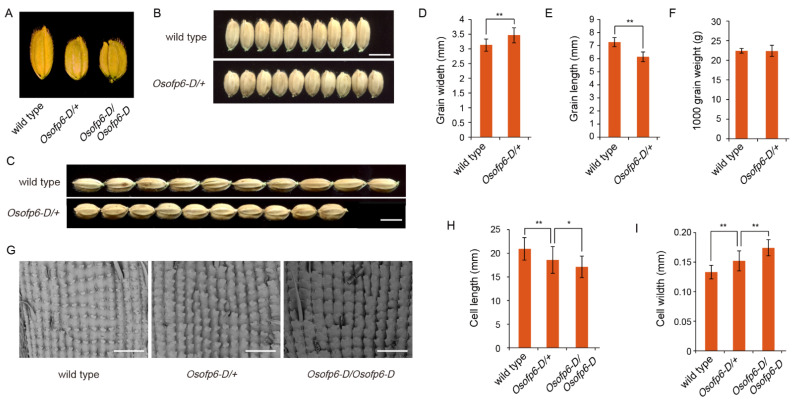
*OsOFP6* overexpression alters the grain shape. (**A**–**C**) Grain phenotypes of the wild-type and *Osofp6-D* mutant. Scale bar: 5 mm. (**D**) Grain width of the wild-type and heterozygous *Osofp6-D* mutant (*Osofp6-D*/+). (**E**) Grain length of the wild-type and heterozygous *Osofp6-D* mutant. (**F**) One thousand grain weight of the wild-type and heterozygous *Osofp6-D* mutant. (**G**) The outer surfaces of the glumes of the wild-type, heterozygous *Osofp6-D* mutant, and homozygous *Osofp6-D* mutant. Scale bar: 500 μm. Cell width (**H**) and cell length (**I**) of the wild-type and *Osofp6-D* mutant. Error bars indicate the means ± SD, *n* = 40. Statistical differences between the wild-type and *Osofp6-D* mutants (*Osofp6-D*/+, *Osofp6-D*/*Osofp6-D*) are indicated by asterisks and were determined by Student’s *t*-test (ns: not significant; * *p* < 0.05; ** *p* < 0.01).

**Figure 4 ijms-25-02889-f004:**
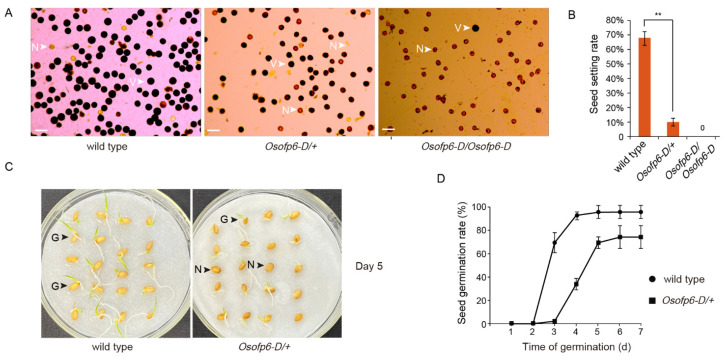
*OsOFP6* overexpression affects the seed setting rate and seed germination. (**A**) Pollen vitality of the wild-type, heterozygous mutants (*Osofp6-D*/+), and homozygous mutants (*Osofp6-D*/*Osofp6-D*). Pollen vitality was observed by a staining test. The florets at the young spike stage of rice were collected, the glume was removed, and the anthers were crushed with tweezers, placed on a slide with I2-KI solution, and then observed with an optical microscope. Scale bar, 100 μm. N: nonviable pollen; V: viable pollen (**B**) Seed setting rate of the wild-type, *Osofp6-D*/+, and *Osofp6-D*/*Osofp6-D*. Error bars indicate the means ± SD, *n* = 15 (** *p* < 0.01). (**C**) Germination of the wild-type and *Osofp6-D*/+ seeds on day 5. G: germinated seed; N: no germinated seed. (**D**) Seed germination rate of the wild-type and *Osofp6-D*/+. Seed germination was scored daily from days 2 to 7. Error bars indicate the means ± SD, *n* = 3. Statistical differences between the wild-type and *Osofp6-D* mutants (*Osofp6-D*/+, *Osofp6-D*/*Osofp6-D*) are indicated by asterisks and were determined by Student’s *t*-test (** *p* < 0.01).

**Figure 5 ijms-25-02889-f005:**
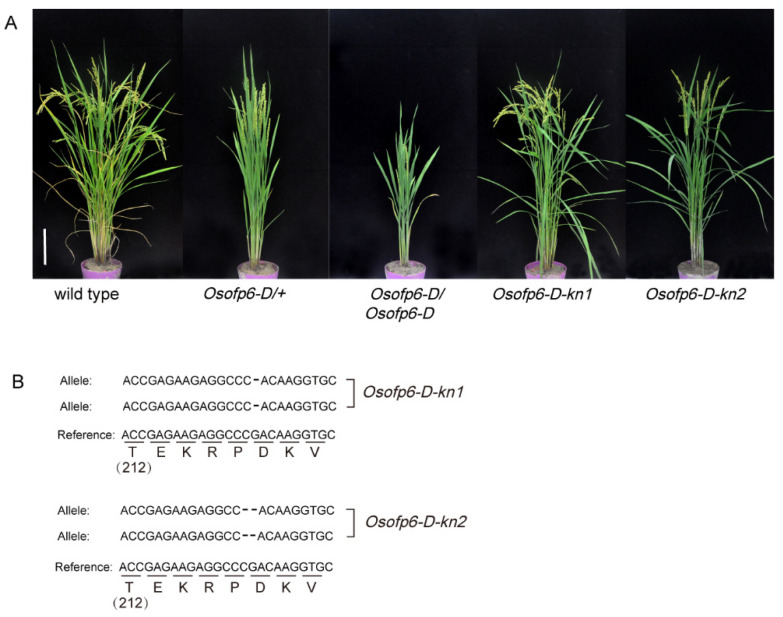
Functional verification of *OsOFP6*. (**A**) Phenotypes of the wild-type, heterozygous mutants (*Osofp6-D*/+), homozygous mutants (*Osofp6-D*/*Osofp6-D*), and *Osofp6-D* knockout plants (*Osofp6-D-kn1*, *Osofp6-D-kn2*). Scale bar: 10 cm. Knockout of the *Osofp6-D* gene with the CRISPR/Cas9 system rescues the *Osofp6-D* phenotype with dwarfism and erect leaves. (**B**) Knockout site of the mutant. Partial nucleotide sequence of the *Osofp6-D-kn1*, *Osofp6-D-kn2*, and wild-type alleles. 212: the amino acid number from the translation start site.

**Figure 6 ijms-25-02889-f006:**
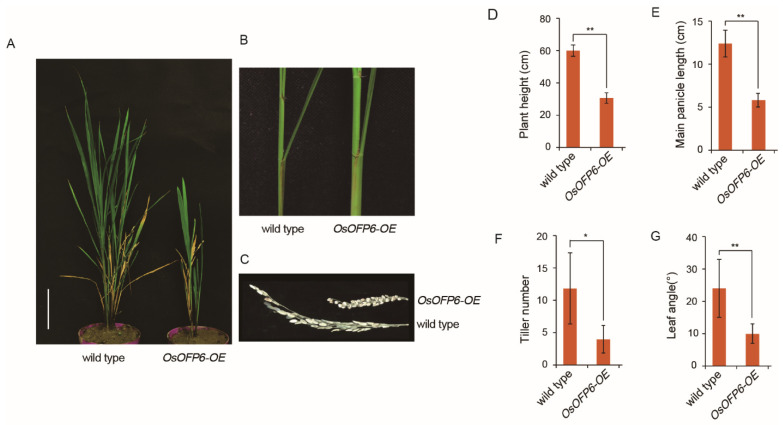
Phenotypes of the wild-type and *OsOFP6-OE*. (**A**) Morphologies of the wild-type (Nipponbare) and *OsOFP6-OE*. Scale bar: 10 cm. (**B**) Morphology of the leaf angle of the wild-type and *OsOFP6-OE*. (**C**) Morphologies of the main panicle at the mature stage of the wild-type and *OsOFP6-OE*. (**D**–**G**) Statistical analysis of plant length (**D**), main panicle length (**E**), tiller number (**F**), and leaf angle (**G**). Error bars indicate means ± SD, *n* = 3. Statistical differences between the wild-type and *OsOFP6-OE* lines are indicated by asterisks and were determined by Student’s *t*-test (* *p* < 0.05; ** *p* < 0.01).

**Figure 7 ijms-25-02889-f007:**
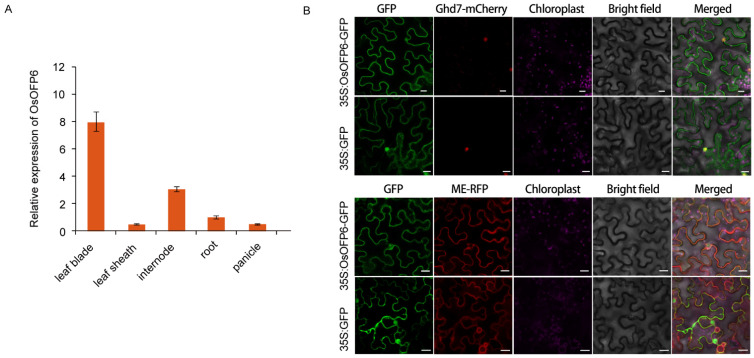
Expression pattern of the *OsOFP6* gene and subcellular location of the protein. (**A**) RT-qPCR analysis of *OsOFP6* expression in various rice tissues in a wild-type background. (**B**) OsOFP6 is localized in the nucleus, as well as the membrane. 35S:OsOFP6-GFP or GFP alone was transiently expressed in *Nicotiana benthamiana* leaves and observed under confocal microscopy. Ghd7-mCherry was the nuclear marker, and ME-RFP was the membrane marker, Scale bar, 20 μm. Error bars indicate means ± SD, *n* = 3.

**Figure 8 ijms-25-02889-f008:**
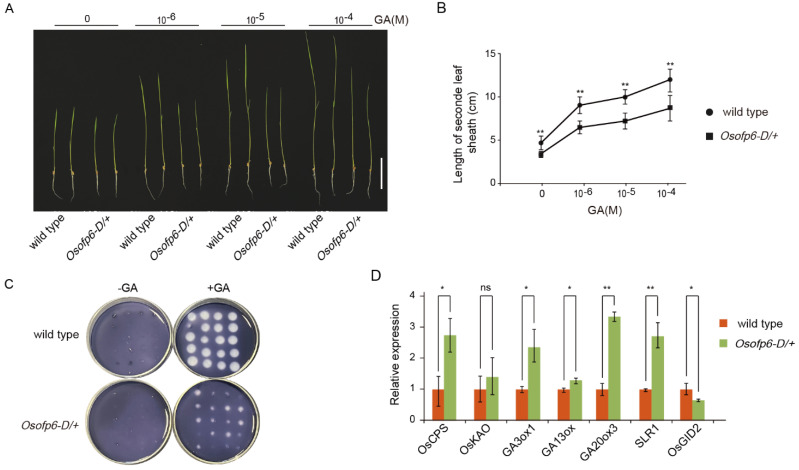
*Osofp6-D* is less sensitive to gibberellin (GA). (**A**) The second leaf sheath elongation of *Osofp6-D* and the wild-type in the presence of the indicated concentration of 10^−6^ M GA3. Scale bar: 5 cm. (**B**) Analysis of the length of the second leaf sheath. (**C**) Plate assay of α-amylase activity induction. Embryo-less half seeds of the wild-type and *Osofp6-D* were incubated on starch plates with or without 10^−6^ M GA3 for 48 h, after which the plates were treated with I_2_-KI solution. This assay was performed three times with similar results. (**D**) Relative expression levels of GA biosynthesis and signaling genes by qRT–PCR. Error bars indicate the means ± SD, *n* = 3. Statistical differences between the wild-type and *Osofp6-D/+* are indicated by asterisks and were determined by Student’s *t*-test (* *p* < 0.05; ** *p* < 0.01, ns, no statistical significance).

**Figure 9 ijms-25-02889-f009:**
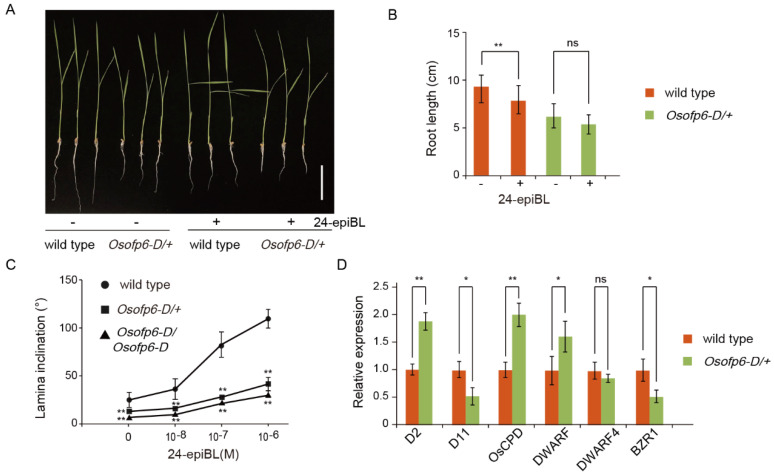
*Osofp6-D* is less sensitive to brassinolide (BR). (**A**) Root growth of *Osofp6-D* and the wild-type in the presence of the indicated concentration of 10^−6^ M 24-epiBL (24-epibrassinolide). Scale bar: 5 cm. (**B**) Statistical analysis of root length. (**C**) Lamina joint bending response to various concentrations of 24-epiBL, as determined by the excised leaf segment method. Seeds were germinated for 2 d, and seedlings were grown for 7 d at 30 °C. Segments comprising part of the second leaf blade, lamina joint, and leaf sheath were incubated in 10^−8^, 10^−7^, or 10^−6^ M 24-epiBL for 48 h in the dark. (**D**) Relative expression levels of BR biosynthesis and signaling genes by qRT–PCR. Error bars indicate means ± SD, *n* = 3. Statistical differences between the wild-type and *Osofp6-D* mutants are indicated by asterisks and were determined by Student’s *t*-test (* *p* < 0.05; ** *p* < 0.01, ns, no statistical significance).

**Figure 10 ijms-25-02889-f010:**
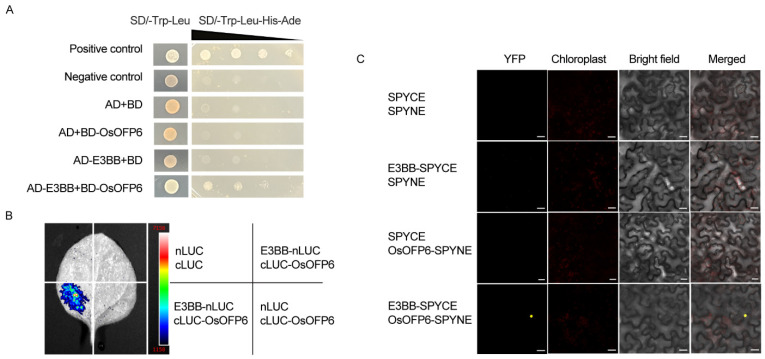
OsOFP6 interacts with E3BB. (**A**) Yeast two-hybrid assay showing that OsOFP6 interacts with E3BB. *OsOFP6* was used as bait (BD), and *E3BB* was used as prey (AD). The left panel shows the growth test on the permissive medium lacking Trp and Leu; the right panel shows the same clones on the selective medium lacking Trp, Leu, His, and Ade. The triangle indicates a 10-fold dilution of yeast. (**B**) LUC complementation imaging (LCI) assays showed that OsOFP6 interacted with E3BB. The full-length coding sequences of *OsOFP6* and *E3BB* were fused with the N-terminal and C-terminal part of the luciferase reporter gene. nLUC and cLUC were used as negative controls. (**C**) Bimolecular fluorescence complementation (BiFC) assays showed that OsOFP6 interacted with E3BB in the epidermal cells of *Nicotiana benthamiana*. OsOFP6 and E3BB were fused with the C and N terminus of YFP, respectively. SPYCE and SPYNE were used as negative controls. YFP reconstruction (yellow dots) was examined when E3BB-SPCYCE and OsOFP6-SPYNE were expressed in *Nicotiana* leaves. Scale bar, 20 μm.

## Data Availability

Data is contained within the article and Appendix A.

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
