# Peer review of "OsOFP6 Overexpression Alters Plant Architecture, Grain Shape, and Seed Fertility"

_ijms, 2024, doi:10.3390/ijms25052889_

Round 1
Reviewer 1 Report (New Reviewer)
Comments and Suggestions for Authors
The document is a detailed research article on the overexpression of OsOFP6 in rice, analyzing its effects on plant architecture, grain shape, and seed fertility. The study discovers that the OsOFP6 mutant exhibited changes like decreased plant height, erect leaves, and altered grain shape due to a T-DNA insertion, leading to increased OsOFP6 expression. Functional verification using CRISPR/Cas9 confirmed these phenotypic changes were due to OsOFP6 overexpression. Additionally, the study explores OsOFP6's interaction with E3BB, suggesting a complex regulatory mechanism impacting rice growth and development, with implications for agricultural practices and crop improvement strategies. The aims of this study is clear and the results are interest to me. In this study, the use of CRISPR/Cas9 technology for functional verification of OsOFP6's role provides strong evidence for the gene's specific impact on rice phenotypes, showcasing the effective application of modern genetic tools in plant research. The interaction between OsOFP6 and E3BB is an important finding, revealing a novel molecular pathway that could be involved in regulating plant growth and development. This interaction suggests a complex network of gene regulation that merits further investigation. I don’t have major comments, but some suggestions I put below:
While the interaction between OsOFP6 and E3BB is established, the study lacks a detailed mechanistic exploration of how this interaction influences plant phenotypes. Understanding the downstream effects of this interaction could provide deeper insights into the biological processes affected. More discussion will be help to solve this point.
The study demonstrates OsOFP6's involvement in GA and BR responses, but the exact role of OsOFP6 within these hormonal pathways is not fully elucidated. Further research is needed to clarify how OsOFP6 modulates these hormone signaling pathways and the implications for plant growth and development. It means the link among OsOFP6, hormone signaling pathways, and plant growth and development should more discussion in this manuscript.
The phenotypic analysis of OsOFP6 overexpression provides valuable insights, yet the study could be strengthened by including more diverse environmental conditions to assess the stability and consistency of the observed traits under different stressors and growth scenarios in the future.
The English writing in this manuscript overall conveys the research findings effectively. However, there are some areas that could be improved for clarity, coherence, and grammatical accuracy.
The manuscript could benefit from a thorough grammar check to correct minor errors and improve readability. For instance, "how OsOFP6 affect rice growth" should be "how OsOFP6 affects rice growth".
Ensure consistent use of scientific terms and abbreviations throughout the document to avoid confusion. For example, if "GA" and "BR" are used initially, they should not be spelled out later unless necessary for clarity.
Some sentences could be restructured for better flow and clarity. For example, the sentence "In addition plants that over- express OsOFP6 were less sensitive to treatment with GA and BR" could be improved for readability.
Attention to detail in capitalization and avoiding typographical errors can enhance the professionalism of the manuscript. For example, the sentence starts with "it provides important reference" should have capitalized "It".
While the summary is concise, adding more specificity about how OsOFP6 interacts with E3BB and affects plant morphology could enrich the understanding for readers not familiar with the topic.
Improving these aspects can enhance the overall quality and readability of the manuscript, making the research findings more accessible and comprehensible to a broader audience.
Comments on the Quality of English LanguageThe manuscript could benefit from a thorough grammar check to correct minor errors and improve readability. For instance, "how OsOFP6 affect rice growth" should be "how OsOFP6 affects rice growth".
Ensure consistent use of scientific terms and abbreviations throughout the document to avoid confusion. For example, if "GA" and "BR" are used initially, they should not be spelled out later unless necessary for clarity.
Some sentences could be restructured for better flow and clarity. For example, the sentence "In addition plants that over- express OsOFP6 were less sensitive to treatment with GA and BR" could be improved for readability.
Attention to detail in capitalization and avoiding typographical errors can enhance the professionalism of the manuscript. For example, the sentence starts with "it provides important reference" should have capitalized "It".
While the summary is concise, adding more specificity about how OsOFP6 interacts with E3BB and affects plant morphology could enrich the understanding for readers not familiar with the topic.
Improving these aspects can enhance the overall quality and readability of the manuscript, making the research findings more accessible and comprehensible to a broader audience.
Author Response
Comments and Suggestions for Authors
The document is a detailed research article on the overexpression of OsOFP6 in rice, analyzing its effects on plant architecture, grain shape, and seed fertility. The study discovers that the OsOFP6 mutant exhibited changes like decreased plant height, erect leaves, and altered grain shape due to a T-DNA insertion, leading to increased OsOFP6 expression. Functional verification using CRISPR/Cas9 confirmed these phenotypic changes were due to OsOFP6 overexpression. Additionally, the study explores OsOFP6's interaction with E3BB, suggesting a complex regulatory mechanism impacting rice growth and development, with implications for agricultural practices and crop improvement strategies. The aims of this study is clear and the results are interest to me. In this study, the use of CRISPR/Cas9 technology for functional verification of OsOFP6's role provides strong evidence for the gene's specific impact on rice phenotypes, showcasing the effective application of modern genetic tools in plant research. The interaction between OsOFP6 and E3BB is an important finding, revealing a novel molecular pathway that could be involved in regulating plant growth and development. This interaction suggests a complex network of gene regulation that merits further investigation. I don’t have major comments, but some suggestions I put below:
Comment:
While the interaction between OsOFP6 and E3BB is established, the study lacks a detailed mechanistic exploration of how this interaction influences plant phenotypes. Understanding the downstream effects of this interaction could provide deeper insights into the biological processes affected. More discussion will be help to solve this point.
Response:
This is a very good and reasonable suggestion. We will conduct additional experiments in a follow-up paper to further investigate how the interaction between OsOFP6 and E3BB affects downstream components, thereby affecting the growth and development of rice. We have also added some content to the discussion section (Line 499, page 13).
Comment:
The study demonstrates OsOFP6's involvement in GA and BR responses, but the exact role of OsOFP6 within these hormonal pathways is not fully elucidated. Further research is needed to clarify how OsOFP6 modulates these hormone signaling pathways and the implications for plant growth and development. It means the link among OsOFP6, hormone signaling pathways, and plant growth and development should more discussion in this manuscript.
Response:
Your suggestion is valid. The additional experiments will be included in a follow-up paper. We have also added some content to the discussion section (Line 468, page 13).
Comment:
The phenotypic analysis of OsOFP6 overexpression provides valuable insights, yet the study could be strengthened by including more diverse environmental conditions to assess the stability and consistency of the observed traits under different stressors and growth scenarios in the future.
Response:
Your suggestion is valid. The additional experiments will be included in a follow-up paper.
Comment:
The English writing in this manuscript overall conveys the research findings effectively. However, there are some areas that could be improved for clarity, coherence, and grammatical accuracy.
The manuscript could benefit from a thorough grammar check to correct minor errors and improve readability. For instance, "how OsOFP6 affect rice growth" should be "how OsOFP6 affects rice growth".
Response:
Thank you very much for finding this error. We are sorry for this grammar problem and have corrected it according to your suggestion (Line 107, page 3). In addition, we also checked the grammar of the entire manuscript.
Comment:
Ensure consistent use of scientific terms and abbreviations throughout the document to avoid confusion. For example, if "GA" and "BR" are used initially, they should not be spelled out later unless necessary for clarity.
Response:
Thank you very much for your suggestion. Except for the first occurrence where the full name was used, all other parts of the entire manuscript have been changed to GA and BR.
Comment:
Some sentences could be restructured for better flow and clarity. For example, the sentence "In addition plants that over- express OsOFP6 were less sensitive to treatment with GA and BR" could be improved for readability.
Response:
Thank you very much for your suggestion. We have made modifications to this sentence. "In addition, plants that overexpress OsOFP6 were less sensitive to treatment with GA and BR" was replaced by “Plants that overexpress OsOFP6 exhibit reduced sensitivity to GA and BR treatments.” (Line 685, page 17)
Comment:
Attention to detail in capitalization and avoiding typographical errors can enhance the professionalism of the manuscript. For example, the sentence starts with "it provides important reference" should have capitalized "It".
Response:
Thank you very much for finding this error. We have corrected it according to your suggestion. (Line 690, page 17)
Comment:
While the summary is concise, adding more specificity about how OsOFP6 interacts with E3BB and affects plant morphology could enrich the understanding for readers not familiar with the topic.
Improving these aspects can enhance the overall quality and readability of the manuscript, making the research findings more accessible and comprehensible to a broader audience.
Response:
Thank you very much for your suggestion. Our study lacks a detailed exploration of the mechanisms by which this interaction affects plant phenotype. We only added a brief introduction to the function of the homologous genes of E3BB in Arabidopsis in the abstract. (Line 24, page 1)
Comments on the Quality of English Language
The manuscript could benefit from a thorough grammar check to correct minor errors and improve readability. For instance, "how OsOFP6 affect rice growth" should be "how OsOFP6 affects rice growth".
Ensure consistent use of scientific terms and abbreviations throughout the document to avoid confusion. For example, if "GA" and "BR" are used initially, they should not be spelled out later unless necessary for clarity.
Some sentences could be restructured for better flow and clarity. For example, the sentence "In addition plants that over- express OsOFP6 were less sensitive to treatment with GA and BR" could be improved for readability.
Attention to detail in capitalization and avoiding typographical errors can enhance the professionalism of the manuscript. For example, the sentence starts with "it provides important reference" should have capitalized "It".
While the summary is concise, adding more specificity about how OsOFP6 interacts with E3BB and affects plant morphology could enrich the understanding for readers not familiar with the topic.
Improving these aspects can enhance the overall quality and readability of the manuscript, making the research findings more accessible and comprehensible to a broader audience.
Response:
These comments are duplicated with the comments above.

Reviewer 2 Report (New Reviewer)
Comments and Suggestions for Authors
Dear Authors,
The paper entitled "Overexpression of OsOFP6 Alters Plant Architecture, Grain Shape, and Seed Fertility" presents results regarding the role of the OsOFP6 gene in regulating the development of rice plants. It contains many interesting materials and new information; however, the paper requires improvement before submission for publication.
Detailed Comments:
1. The Abstract is well-structured.
2. The Introduction adequately introduces the research topic; however, the end of the Introduction should clearly explain the research aim or propose an alternative research hypothesis in addition to the null hypothesis. The conclusions from the study should be placed in the final chapter.
3. The "Materials and Methods" section is sufficiently described. The authors have detailed the plant materials used, experimental techniques, and laboratory protocols, enabling the reproducibility of the experiments conducted. The data analysis method has also been described, which is crucial for understanding the results of the study. This allows readers to gain a comprehensive understanding of the research process and the potential limitations and advantages of the experiments conducted.
4. Results: The authors demonstrated that the overexpression of the OsOFP6 gene influences plant architecture, grain shape, and seed fertility. Phenotypic studies showed that plants overexpressing OsOFP6 exhibited altered morphological traits, such as plant height, panicle length, tiller number, grain length, grain weight, and seed setting rate. Additionally, experiments conducted showed that OsOFP6 influences plant hormonal responses by regulating genes associated with gibberellic acid (GA) and brassinosteroid (BR) biosynthesis. Interactions between the OsOFP6 gene and the E3BB protein suggest the role of this gene in controlling plant development by regulating protein activity.
5. The Discussion was conducted in a detailed and understandable manner. The authors discussed the results obtained in the context of existing scientific knowledge and presented interpretations and conclusions drawn from the research conducted. Potential implications and significance of the results for further research on the topic and for agricultural practice were also discussed. At the same time, the authors considered the limitations of their work and proposed potential directions for further research, indicating a comprehensive approach to analyzing the results and their interpretation.
6. The conclusions of the study indicate the significant importance of the OsOFP6 gene in regulating plant architecture, grain shape, and seed fertility in rice plants. Additional functional studies may further elucidate the molecular mechanisms involved in these processes. However, this should be emphasized, and conclusions should be formulated in a separate chapter after discussing the results.
Strengths:
1. Diverse research methods: The paper utilizes various research methods, such as phenotypic analysis, subcellular localization studies, seed germination experiments, and biochemical and molecular experiments, contributing to a comprehensive understanding of the function of the OsOFP6 gene in rice plants.
2. In vivo and in vitro experiments: The authors conducted experiments on both live plants and cell cultures, allowing verification of the results at different levels of biological organization.
3. Conclusions based on phenotypic and molecular observations: The paper combines phenotypic observations with gene expression analysis and protein activity, allowing inference about the molecular mechanisms regulating plant development.
4. Use of diverse microscopy techniques: The authors employed light and confocal microscopy as well as electron microscopy, enabling precise examination of cellular and subcellular protein localization.
Weaknesses:
1. Lack of full statistical data analysis: Some experimental results are presented without a full statistical analysis, limiting the credibility of the conclusions.
2. Need for further functional studies: Although the paper identifies the role of the OsOFP6 gene in regulating the development of rice plants, further functional studies are needed to better understand the molecular mechanisms underlying this process.
3. The paper does not discuss potential practical implications of the results for agriculture, which may limit their practical application.

Minor English editing is required
Author Response
The paper entitled "Overexpression of OsOFP6 Alters Plant Architecture, Grain Shape, and Seed Fertility" presents results regarding the role of the OsOFP6 gene in regulating the development of rice plants. It contains many interesting materials and new information; however, the paper requires improvement before submission for publication.
Detailed Comments:
- The Abstract is well-structured.
Comment:
- The Introduction adequately introduces the research topic; however, the end of the Introduction should clearly explain the research aim or propose an alternative research hypothesis in addition to the null hypothesis. The conclusions from the study should be placed in the final chapter.
Response:
Thank you very much for your suggestion. These sentences were rephrased according to your comment. (Line 105, page 3) (Line 684, page 17)
- The "Materials and Methods" section is sufficiently described. The authors have detailed the plant materials used, experimental techniques, and laboratory protocols, enabling the reproducibility of the experiments conducted. The data analysis method has also been described, which is crucial for understanding the results of the study. This allows readers to gain a comprehensive understanding of the research process and the potential limitations and advantages of the experiments conducted.
- Results: The authors demonstrated that the overexpression of the OsOFP6 gene influences plant architecture, grain shape, and seed fertility. Phenotypic studies showed that plants overexpressing OsOFP6 exhibited altered morphological traits, such as plant height, panicle length, tiller number, grain length, grain weight, and seed setting rate. Additionally, experiments conducted showed that OsOFP6 influences plant hormonal responses by regulating genes associated with gibberellic acid (GA) and brassinosteroid (BR) biosynthesis. Interactions between the OsOFP6 gene and the E3BB protein suggest the role of this gene in controlling plant development by regulating protein activity.
- The Discussion was conducted in a detailed and understandable manner. The authors discussed the results obtained in the context of existing scientific knowledge and presented interpretations and conclusions drawn from the research conducted. Potential implications and significance of the results for further research on the topic and for agricultural practice were also discussed. At the same time, the authors considered the limitations of their work and proposed potential directions for further research, indicating a comprehensive approach to analyzing the results and their interpretation.
Comment:
- The conclusions of the study indicate the significant importance of the OsOFP6 gene in regulating plant architecture, grain shape, and seed fertility in rice plants. Additional functional studies may further elucidate the molecular mechanisms involved in these processes. However, this should be emphasized, and conclusions should be formulated in a separate chapter after discussing the results.
Response:
Thank you very much for your suggestion. We have added the conclusion section. (Line 684, page 17)
Strengths:
- Diverse research methods: The paper utilizes various research methods, such as phenotypic analysis, subcellular localization studies, seed germination experiments, and biochemical and molecular experiments, contributing to a comprehensive understanding of the function of the OsOFP6 gene in rice plants.
- In vivo and in vitro experiments: The authors conducted experiments on both live plants and cell cultures, allowing verification of the results at different levels of biological organization.
- Conclusions based on phenotypic and molecular observations: The paper combines phenotypic observations with gene expression analysis and protein activity, allowing inference about the molecular mechanisms regulating plant development.
- Use of diverse microscopy techniques: The authors employed light and confocal microscopy as well as electron microscopy, enabling precise examination of cellular and subcellular protein localization.
Weaknesses:
Comment:
- Lack of full statistical data analysis: Some experimental results are presented without a full statistical analysis, limiting the credibility of the conclusions.
Response:
Thank you for underlining this deficiency. We have added the statistical analysis results for Figure 8B (Line 328, page 9) and Figure 9C (Line 359, page 10).
Comment:
- Need for further functional studies: Although the paper identifies the role of the OsOFP6 gene in regulating the development of rice plants, further functional studies are needed to better understand the molecular mechanisms underlying this process.
Response:
Thank you very much for your suggestion. The additional experiments will be included in a follow-up paper.
Comment:
- The paper does not discuss potential practical implications of the results for agriculture, which may limit their practical application.
Response:
This is a very good suggestion. We believe that there is still a gap between the current research results and practical applications.

This manuscript is a resubmission of an earlier submission. The following is a list of the peer review reports and author responses from that submission.
Round 1
Reviewer 1 Report
Comments and Suggestions for Authors
Dear Authors,
I had great opportunity to review the article entitled „ OsOFP6 overexpression alters plant architecture, grain shape, and seed fertility” which is considered for IJMS Journal. The work present new insight in OVATE family proteins. The authors performed a lot of interesting analyzes but the clarity of its presentations and also used nomenclature is not appropriate.
1. Introduction section
This part must have precisely formulated aim or hypothesis of the study according IJMS Journal publication rules. Authors must add precisely formulated aim of the study because now nothing like that is not present.
Results section
First of all figures are overloaded with data which make them extremely small and not acceptable quality for IJMS. Authors must analyze all Figures to select results and arrange them in logical manner. Problems with specific Figures:
Figure 1 Part C is unreadable because of size and quality
Figure 2 authors must characterize the phenotype of plants mark important differences of figures. The statement phenotype and number of photo is completely illogical.
Figure 3. Parts A B C and G are extremely low quality
Figure 4 parts A and C has no marking showing differences and they are low quality
Figure 5 part A has the same problem as Figure 2
Figure 7 Part B and Figure 10 part C did not show subcellular localization it show localization in confocal. For subcellular localizations authors must performed TEM with magnification between 4000x-22 000x not on level of confocal
Figure 8 and 9 overloaded and low quality
Materials and Methods Section
This is very problematic. How It is even possible that authors written this section with use 4 position of literature references. This suggest that they developed all of methods in their laboratory which is simply not true. This suggest the serious ethical/plagiarism problem. Moreover where authors did not present methodology for part G of Figure no information about preparation for SEM
Sincerely,
Author Response
Dear Editors and Reviewers:
Please download the revised version of our manuscript “OsOFP6 overexpression alters plant architecture, grain shape, and seed fertility” (Manuscript ID: ijms-2687788). We would like to thank the anonymous reviewers for helpful comments and suggestions on the previous version of this manuscript. We believe that our work has benefited substantially from the invaluable input of the review team. Below is the detail of how we have addressed the critique of the review team.
Response to reviewer 1
Comment: I had great opportunity to review the article entitled “OsOFP6 overexpression alters plant architecture, grain shape, and seed fertility” which is considered for IJMS Journal. The work present new insight in OVATE family proteins. The authors performed a lot of interesting analyzes but the clarity of its presentations and also used nomenclature is not appropriate.
Response: We apologize for not expressing ourselves clearly. In the revised manuscript, some changes have been made to make the expression clearer and the semi-dominant mutant was renamed Osofp6-D. If there are other errors or further requests, please contact us.
Comment: 1. Introduction section
This part must have precisely formulated aim or hypothesis of the study according IJMS Journal publication rules. Authors must add precisely formulated aim of the study because now nothing like that is not present.
Response: We supplemented the aim of the article in the last paragraph of the introduction section.
Comment: Results section
First of all figures are overloaded with data which make them extremely small and not acceptable quality for IJMS. Authors must analyze all Figures to select results and arrange them in logical manner. Problems with specific Figures:.
Response: We don't think our figures are overloaded. The number of photos or histograms in each of our figures is similar to the number of histograms or photos in other published articles. Here are some examples:
- Gao, Q.; Yin, X.; Wang, F.; Hu, S.; Liu, W.; Chen, L.; Dai, X.; Liang, M. OsJRL40, a Jacalin-Related Lectin Gene, Promotes Salt Stress Tolerance in Rice. Int. J. Mol. Sci. 2023, 24, doi:10.3390/ijms24087441.
- Hu, D.; Li, M.; Zhao, F.-J.; Huang, X.-Y. The Vacuolar Molybdate Transporter OsMOT1;2 Controls Molybdenum Remobilization in Rice. Front. Plant Sci. 2022, 13, 863816, doi:10.3389/fpls.2022.863816.
Comment: Figure 1 Part C is unreadable because of size and quality
Response: the readability of Figure 1 Part C has been improved in the revised manuscript.
Comment: Figure 2 authors must characterize the phenotype of plants mark important differences of figures. The statement phenotype and number of photo is completely illogical.
Response: We have made modifications and added a description of the main panicle length in the revised manuscript. We believe that the histograms clearly show the differences between the mutants and wild type.
Comment: Figure 3. Parts A B C and G are extremely low quality
Response: Parts A, B, C, and G were updated. These images are clear enough to identify the objective traits.
Comment: Figure 4 parts A and C has no marking showing differences and they are low quality
Response: We have added new annotations to the captions and photos in the revised manuscript. These images are clear enough to identify the objective traits.
Comment: Figure 5 part A has the same problem as Figure 2
Response: We have added new annotations to the captions.
Comment: Figure 7 Part B and Figure 10 part C did not show subcellular localization it show localization in confocal. For subcellular localizations authors must performed TEM with magnification between 4000x-22 000x not on level of confocal
Response: Subcellular localization using confocal microscopy is a commonly used method in plant science. Please refer to the following literature:
- Gao, Q.; Yin, X.; Wang, F.; Hu, S.; Liu, W.; Chen, L.; Dai, X.; Liang, M. OsJRL40, a Jacalin-Related Lectin Gene, Promotes Salt Stress Tolerance in Rice. Int. J. Mol. Sci. 2023, 24, doi:10.3390/ijms24087441.
- He, Y.; Duan, W.; Xue, B.; Cong, X.; Sun, P.; Hou, X.; Liang, Y.-K. OsαCA1 Affects Photosynthesis, Yield Potential, and Water Use Efficiency in Rice. Int. J. Mol. Sci. 2023, 24, doi:10.3390/ijms24065560.
Comment: Figure 8 and 9 overloaded and low quality.
Response: We have made some improvements in the new version.
Comment: Materials and Methods Section
This is very problematic. How It is even possible that authors written this section with use 4 position of literature references. This suggest that they developed all of methods in their laboratory which is simply not true. This suggest the serious ethical/plagiarism problem. Moreover where authors did not present methodology for part G of Figure no information about preparation for SEM.
Response: In the first version, we cited 9 references in the methodology section (including References 6, 16, 27-33). In the revised version, we added 4 new references, including a reference for SEM, in the methodology section.

Reviewer 2 Report
Comments and Suggestions for Authors
The article entitled "Unraveling the Role of OsOFP6 in Rice Growth and Development: Insights from a Gain-of-Function Mutant" explores the previously unknown functions of OVATE family proteins (OFPs) in rice, focusing on OsOFP6. The study identifies a novel gain-of-function rice mutant, ofp6-D, which displays a range of distinctive phenotypic changes, including decreased plant height, erect leaves, reduced panicle size, short and wide seeds, delayed seed germination, and reduced fertility. These changes are attributed to the increased expression of OsOFP6, resulting from a T-DNA insertion. The researchers confirm the causal relationship between the observed phenotypes and OsOFP6 overexpression by complementing the ofp6-D phenotype through the knockout of OsOFP6 using the CRISPR/Cas9 system. Overall, this study advances our knowledge of the OFP family's role in shaping the physical characteristics of rice plants and opens avenues for further research in crop improvement strategies. In conclusion, the paper would be sufficient to merit publication in JPR, though a revision is recommended which needs to include the following points.
(1) Introduction and discussion
The results of the characterization of the OsOFP6 OE and KO seem to differ from those previously made by Ma et al. Possible reasons should be discussed in the discussion.
(2) Results, T-DNA insertion mutant
From the results of flanking sequence analysis, can the authors say that the number of T-DNA insertions is one or very few? If the authors can say so, then describing it will increase the reliability of the results.
(3) 2.1 T-DNA insertion co-segregates with the ofp6-D mutant
The authors should indicate how many F2 plants genotyped to show the co-segregation.
(4) Supplementary Figure 2 should be one of the main figure (may be part of Figure 1). This result is very important for this paper.
(5) line 233-235
No figures or references to support the description.
(6) The use of the CRISPR/Cas9 system to complement the ofp6-D phenotype by knocking out OsOFP6 is a standard approach. However, the study does not thoroughly discuss potential off-target effects or unintended genetic modifications that could influence the observed outcomes. Some explanation on this issue should be given.
Author Response
Dear Editors and Reviewers:
Please download the revised version of our manuscript “OsOFP6 overexpression alters plant architecture, grain shape, and seed fertility” (Manuscript ID: ijms-2687788). We would like to thank the anonymous reviewers for helpful comments and suggestions on the previous version of this manuscript. We believe that our work has benefited substantially from the invaluable input of the review team. Below is the detail of how we have addressed the critique of the review team.
Response to reviewer 2
Comment: (1) Introduction and discussion
The results of the characterization of the OsOFP6 OE and KO seem to differ from those previously made by Ma et al. Possible reasons should be discussed in the discussion.
Response: Ma et al. only reported on the OsOFP6 RNAi plants but not on OsOFP6 KO plants. RNAi may also interfere with other genes in addition to interfering with OsOFP6. We are also very confused about the phenotypic differences between our OsOFP6 overexpressing plants and their OsOFP6 overexpressing plants. Based on our evidence, we believe that our results are more reliable. The relevant discussion has been placed in the second paragraph of the discussion section.
Comment: (2) Results, T-DNA insertion mutant
From the results of flanking sequence analysis, can the authors say that the number of T-DNA insertions is one or very few? If the authors can say so, then describing it will increase the reliability of the results.
Response: The results of flanking sequence analysis cannot be used for analyzing copy numbers of T-DNA insertions as some insertion sites may be omitted using TAIL-PCR. There may only be one copy for T-DNA insertions in the transgenic line with the Osofp6-D mutant because no hygromycin resistance gene, which is within the T-DNA region, was amplified in wild-type plants of the transgenic line (Supplementary Figure 1).
Comment: (3) 2.1 T-DNA insertion co-segregates with the ofp6-D mutant
The authors should indicate how many F2 plants genotyped to show the co-segregation.
Response: Your suggestion is valid. Sixty F2 generation plants, including 20 wild type, heterozygous, and homozygous mutants each were used for co-segregation analysis. We have added this information in the new version.
Comment: (4) Supplementary Figure 2 should be one of the main figure (may be part of Figure 1). This result is very important for this paper.
Response: Thank you for your suggestion. The Supplementary Figure 2 has been placed in the Figure 1B.
Comment: (5) line 233-235
No figures or references to support the description.
Response: Thank you for your suggestion. We have added two references (references 1 and 9) and Figure 2 to the revised version.
Comment: (6) The use of the CRISPR/Cas9 system to complement the ofp6-D phenotype by knocking out OsOFP6 is a standard approach. However, the study does not thoroughly discuss potential off-target effects or unintended genetic modifications that could influence the observed outcomes. Some explanation on this issue should be given.
Response: That is a very good and reasonable question. We believe that the phenomenon of CRISPR off-target is not related to this experiment. CRISPR off-target is nothing more than the creation of new mutation sites. In fact, the transgenic process for introducing CRISPR system itself also results in many mutations. We obtained multiple independent events that restored the Osofp6-D phenotype through KO of Osofp6-D. In addition, plants overexpressing OsOFP6 with the 35S enhancer mimicked the Osofp6-D phenotypes. Knocking out the OsOFP6 gene in wild-type plants does not produce a visible mutant phenotype. These results strongly demonstrate the correlation between the Osofp6-D phenotype and the OsOFP6 gene.
Round 2
Reviewer 1 Report
Comments and Suggestions for Authors
Dear Authors,
Problems which still persist
" Figures are overloaded with data which make them extremely small and not acceptable quality for IJMS. The a lot different elements lover quality of almost all results. Again Authors must analyze all Figures to select results and arrange them Moreover still Figure 2 authors must characterize the phenotype of plants mark important differences of figures. The statement phenotype and number of photo is completely illogical.
“Figure 7 Part B and Figure 10 part C did not show subcellular localization it show localization in confocal. For subcellular localizations authors must performed TEM with magnification between 4000x-22 000x not on level of confocal” Response of authors is strange the subcellular means that you are able t know exactly where in specific organelle you are able to spot elements which you localized for example in thylakoids in plastids. The confocal simply did not give you enough magnification. So adding the other publication in response suggest that they copy the error or other authors.
Sincerely,